physical chemistry/inorganic chemistry

pyroelectric, spin-flop, ferroelectric, single crystal

**Author for correspondence:**
Kaige Gao
e-mail: kggao@yzu.edu.cn

This article has been edited by the Royal Society of Chemistry, including the commissioning, peer review process and editorial aspects up to the point of acceptance.

# Pyroelectricity and field-induced spin-flop in (4-(Aminomethyl)pyridinium)₂ MnCl₄ · 2H₂O

Kaige Gao, Chunlin Liu, Wei Zhang, Kangni Wang and Wenlong Liu

College of Physical Science and Technology, Yangzhou University, Jiangsu 225009, People's Republic of China

 KG, 0000-0002-7634-3400

Large single crystals of (4-(Aminomethyl)pyridinium)₂ MnCl₄ · 2H₂O (**1**) were grown by slow evaporation of solution. The crystal structure was solved to be P1̄, which belongs to the central symmetric space group. But small pyroelectric current was detected, as well as a ferroelectric hysteresis loop. The pyroelectric and the ferroelectric properties were attributed to the strain caused by defects. Temperature-dependent magnetic curves and the *M–H* curve show that **1** is antiferromagnetic ordering below 2.5 K. A field-induced spin-flop is observed in the antiferromagnetic ordering state.

## 1. Introduction

Organic–inorganic hybrid perovskites have attracted great attention due to their rich and excellent physical properties [1–5]. In hybrid perovskites crystals, BX₆ octahedra play an important role in their multifunction property. Many ferroelectric materials have been found among organic–inorganic hybrid perovskite materials [4,6–19]. The octahedral structure is important for the ferroelectricity of hybrid perovskites. Some of the organic–inorganic hybrid perovskites are found to possess large spontaneous polarization and large piezoelectric coefficient that is comparable to those of inorganic materials. The excellent ferroelectric property and piezoelectric property of organic–inorganic perovskites make molecular-based ferroelectrics become attractive in the field of application [20–22]. Here, we report a compound (4-(Aminomethyl)pyridinium)₂ MnCl₄ · 2H₂O which has organic–inorganic hybrid perovskite structure. Pyroelectricity and field-induced spin-flop were found in the single crystal of compound **1**.

Flexoelectricity reflects a coupling between polarization and strain [23–26]. Most materials have flexoelectric effect, in which

a strain gradient could induce a polarization and a piezoelectric composite containing no piezoelectric elements. In centrosymmetric materials, a strain gradient can break the inversion symmetry. The strain-induced polarization in $SrTiO_3$ is widely studied. The crystal structure of $SrTiO_3$ belongs to the simple cubic centrosymmetric lattices, and the dielectric permittivity of $SrTiO_3$ is large. The $ABX_3$ crystal structure and the large dielectric permittivities make it easy for strain to induce a polarization in the $SrTiO_3$ single crystal [27,28]. The bulk photovoltaic effect can also exist in $SrTiO_3$, which is caused by flexoelectricity [26]. Pyroelectricity in **1** is similar that found in $SrTiO_3$ [29], which is not caused by the ferroelectricity or spontaneous polarization. The space group of a single crystal of **1** is P$\bar{1}$, which is centrosymmetric symmetry. However, the small pyroelectric current is detected in **1** using Chynoweth technology. The flexoelectric effect may play an important role in the pyroelectric effect in **1**.

# 2. Material and methods

## 2.1. Synthesis

4-(Aminomethyl)pyridine, HCl and $MnCl_2$ were added in water with the ratio of 2 : 2.1 : 1 to form a clear colourless solution. Large single crystals of compound **1** were obtained by slow evaporation of the clear solution.

## 2.2. Measurements

The single-crystal X-ray diffraction was performed on a Bruker D8 QUEST at room temperature. The powder X-ray diffraction (PXRD) was measured on a Rigaku D/MAX 2000 PC X-ray diffractometer. Differential scanning calorimetry (DSC) measurements of single crystals were recorded by a NETZSCH DSC 200F3. The DSC of **1** was measured from 120 to 470 K with sweeping speed of 10 K min$^{-1}$. $P–E$ hysteresis loops were recorded on a Precision Premier II (Radiant Technologies, Inc.). The $P–E$ hysteresis loop of **1** was measured on the single crystals at room temperature with frequency of 50 Hz. The dynamic pyroelectric current was measured using the Chynoweth technique. The current was recorded with an electrometer (Keithley 6517B). A 1047 nm pulsed laser with a power of 100 mW was used to heat the sample. Magnetic measurements of the samples were performed on a Quantum Design SQUID (MPMS XL) magnetometer. The magnetic susceptibility of **1** was measured from 300 to 2 K with a speed of 5 K min$^{-1}$.

# 3. Results and discussion

To determine the crystal structure of **1**, single-crystal X-ray diffraction was performed at room temperature. The crystal structure was solved with Shelxtl program. When solving the crystal structure with Xprep, P1 and P$\bar{1}$ space group were advised with closed combined figure of merit (CFOM) value. It is hard to determine the space group. When solved with P1 space group, the flack factor is 0.35. However, checkcif suggests P$\bar{1}$ space group. So the final structure was resolved with P$\bar{1}$. Figure 1 shows the crystal structure of **1** at room temperature. There are six atoms around the Mn atom, four chlorine atoms, and two oxygen atoms. The two oxygen atoms come from water molecules. Of the four chlorine atoms, two come from $MnCl_2$, and the other two are from hydrochloric acid. Mn is in the centre of the octahedron constructed by four chlorines and two oxygens, which may contribute greatly to polarization. Hydrochloric acid only protonates a N on the 4-(Aminomethyl)pyridine. The nitrogen on the amine group is protonated and the nitrogen on the pyridine ring is not protonated. The water molecules form hydrogen bonds with N on the pyridine ring. Hydrogen bonds are helpful for the polarization to switch under an applied electric field. A pair of 4-(Aminomethyl)pyridine is antiparallel, making the crystal structure centrosymmetric. Powder XRD was performed, which corresponds well with the pattern simulated from the single crystal structure (figure 1b). Amorphization occurs during the grinding process, and a large amorphous package of powder XRD appears around 20°. Even after grinding for a few minutes, amorphization still exists, indicating that the crystal structure is sensitive to pressure.

DSC measurements were performed from 123 to 473 K (electronic supplementary material figure S1), which is sensitive to phase transitions. No reversible phase transition was found by DSC measurements. Only two exothermic peaks centred at 405 and 425 K appear during the heating process. But no anomaly was found on the DSC curve during the cooling process. The heat capacity corresponding to these two endothermic peaks is large, indicating that these two peaks are dehydration peaks. The two exothermic peak tell that compound **1** lost its two crystal waters at 405 and 425 K. The dehydrated

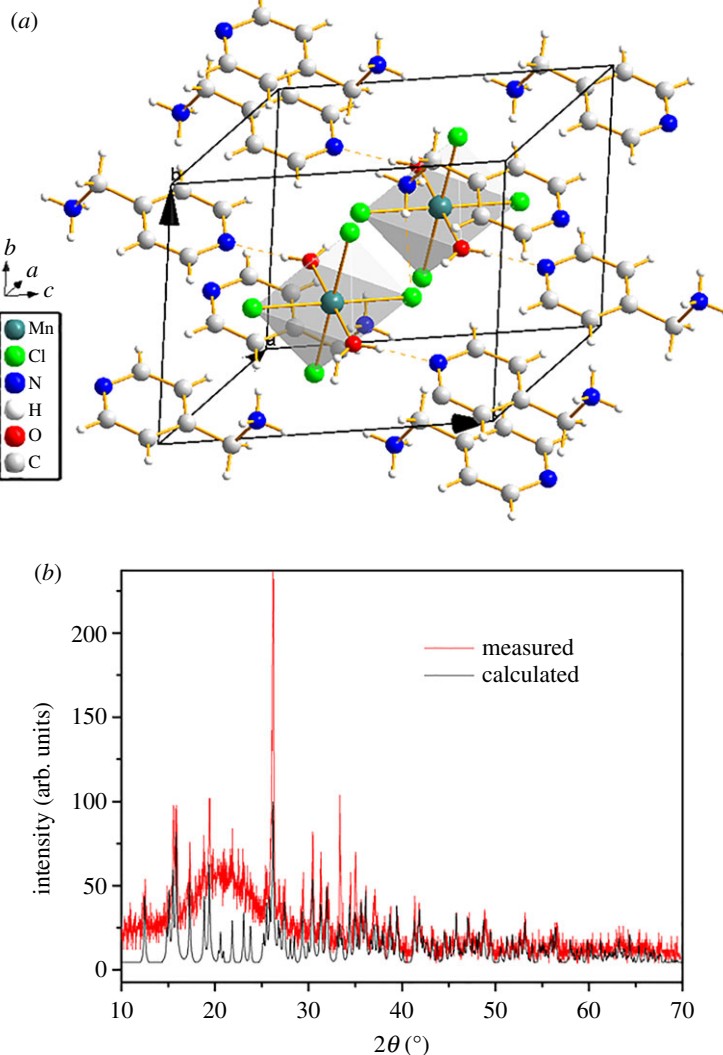

**Figure 1.** (*a*) The crystal structure of **1** at room temperature. (*b*) Powder XRD pattern of the compound **1** compared with the calculated PXRD pattern deduced from the crystal structure.

temperature of the compound is high due to the coordination bond between water molecules and manganese atoms. The coordination bonds make the crystals of compound **1** stable below 370 K. No anomaly peak appears in the second cycle of DSC measurements indicating that the dehydrated sample does not have phase transitions.

The pyroelectric current was measured with Chynoweth method (figure 2). Chynoweth method is a kind of period pulse pyroelectric measurement, which is sensitive to pyroelectric current. A 1047 nm laser was used as the heating source. The laser was turned on and off. Figure 2 shows the pyroelectric current measured on the (021) face of single crystal of **1** with a period of 50 s. When the laser was turned on, the pyroelectric current of **1** decreased and then increased slowly. However, once the laser was turned off, the current increased sharply and then decreased slowly. As the base line of the current is not zero, which exists before the measurements, the direction of the measured current is not reversed. The base current usually comes from the conductivity of the sample. The pyroelectric effect usually exists in polar crystal classes. All polar crystals are pyroelectric. But single-crystal X-ray diffraction suggests a P1̄ space group, which is centrosymmetric. There is no pyroelectric effect in the crystals of the central symmetric space group. The pyroelectric current of compound **1** proves polar composition existing in the crystal. The pyroelectric current of compound **1** is small, only 0.2 pA for an area of 10 mm². So the pyroelectric coefficient is small in the compound. The pyroelectric current may originate from the polar composition induced by the strain, which may be induced by defects or uneven thermal expansion and contraction. When the laser is turned on and off, uneven heating and cooling of the large single crystal causes stress to form in the crystal. The stress causes a flexoelectric effect which can produce polar composition in the compound.

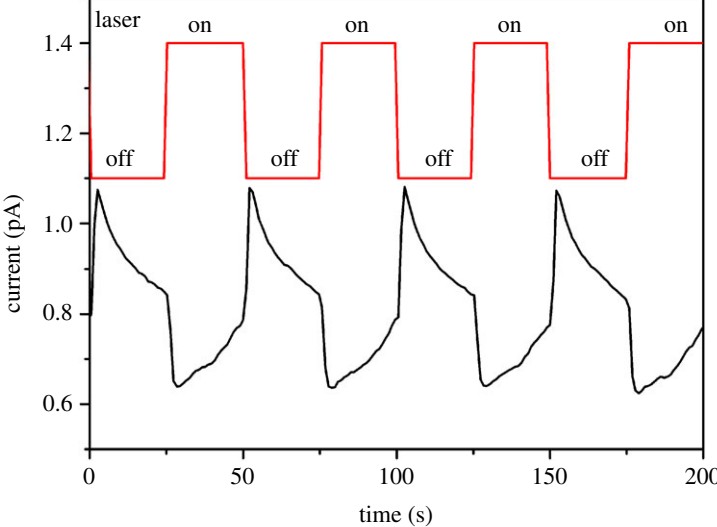

**Figure 2.** Pyroelectric current measured with Chynoweth technique.

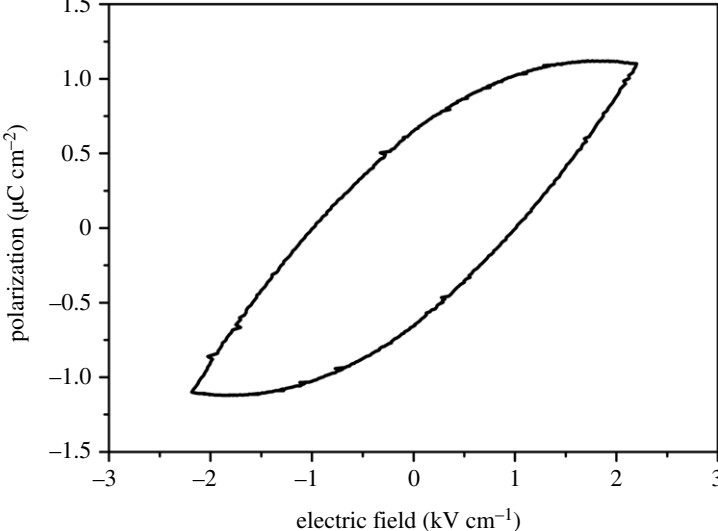

**Figure 3.** The ferroelectric hysteresis loop of compound **1**.

On the other hand, the single-crystal X-ray diffraction result shows that the flack factor of the compound is 0.35 when solved in P1. The flack factor is larger than ordinary polar crystal but smaller than Centro symmetric crystal. It is difficult to determine the absolute configuration of **1**. But Centro symmetric P$\bar{1}$ seems more reliable than P1 according to the checkcif result. Although the crystal structure was solved in P$\bar{1}$ space group, the flack factor obtained from P1 may suggest polar composition in the single crystal of compound **1**. Defects may bring polar composition in the crystal. Accompanied with laser on and off, stain gradient induced by uneven heating and cooling can increase the polar composition, which makes the pyroelectric current detectable.

The polarization versus electric field hysteresis loop was acquired using (021) face of the single crystal (figure 3). A ferroelectric hysteresis loop was observed. But the hysteresis loop shows that the polarization is not fully saturated. The polarization cannot be fully saturated even when the electric field reaches breakdown electric field. It is known that electrostriction effect exists in all dielectric materials [30]. The strain is proportional to the square of the polarization under the electrostriction effect. So strain appeared when the electric field was taken on the single crystal of the compound. It has been proposed that polar composition may exist in the single crystal of the compound according to the pyroelectric measurements and single-crystal XRD measurements. The strain originating from the electrostriction effect could also increase the polar composition. The additional strain would bring strain gradient around defects in the crystal, which could produce polar composition due to the flexoelectric effect in turn.

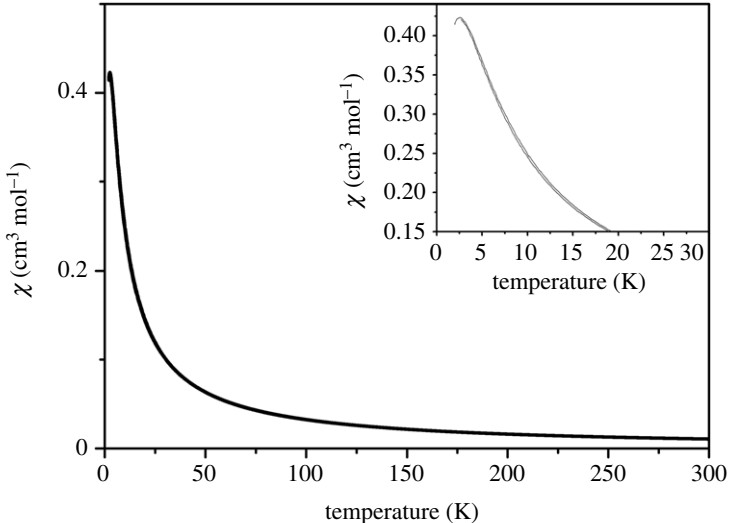

**Figure 4.** The temperature-dependent magnetic susceptibility.

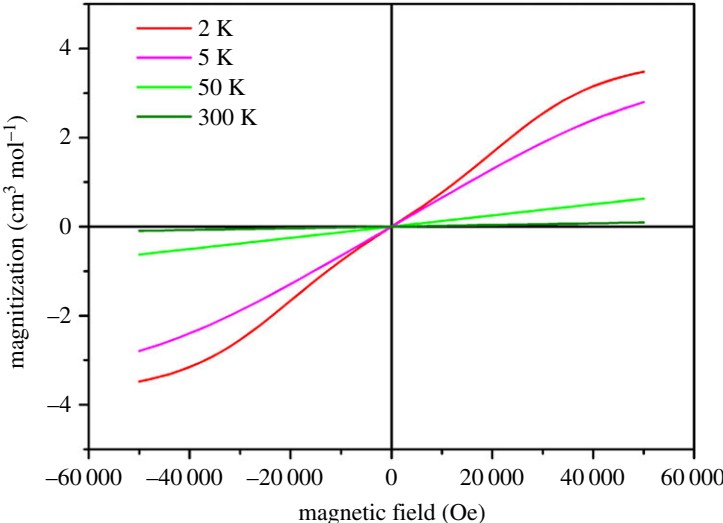

**Figure 5.** $M–H$ curve measured at 2, 5, 50 and 300 K.

The ferroelectric and pyroelectric behaviour in centrosymmetric compound **1** is similar to that in $SrTiO_3$. $SrTiO_3$ is a famous centrosymmetric crystal material. Strain-induced ferroelectric has long been found existing in $SrTiO_3$ [31–33]. Many methods have been developed to introduce stain in $SrTiO_3$ in order to destroy the central symmetry [34]. As the structure of $SrTiO_3$ is close to the polar structure, it is easy for strain to introduce polar composition in the crystal. The flexoelectric effect of $SrTiO_3$ is larger than most centrosymmetric materials. The polar structure and the non-polar structure of **1** are very close, so stress and defects can make the polar component stable. In addition, it is easier to introduce polar composition in **1** than in $SrTiO_3$, because the molecule in **1** makes it more flexible than inorganic $SrTiO_3$.

Multifunctional magnetic materials have caught increasing attention recently due to the possibility of producing materials where the magnetism can be modified by tuning the others [35–38]. The temperature-dependent molar magnetic susceptibility $\chi(T)$ of compound **1** measured at $H = 5000$ Oe is presented in figure 4. With decreasing temperature, $\chi(T)$ increases and shows a round peak around 2.5 K ($T_N$). Above 7 K, the magnetic susceptibility obeys Curie–Weiss law ($\chi = C/(T - T_c)$) with $T_c = -2.45$ K (electronic supplementary material, figure S2). Considering the negative Weiss temperature, the anomaly marks the antiferromagnetic ordering below $T_N$.

$M–H$ loops were measured at 300, 50, 5 and 2 K (figure 5). No hysteresis loops were found for all the $M–H$ curves. A field-induced spin-flop transition is observed in the antiferromagnetic ordering phase. In

the AFM ordering state, $M(H)$ displays a linear field dependence when $H < 2000$ Oe, but undergoes a weak step-like increase around $20\,000$ Oe. This anomaly, which is ascribed to a field-induced metamagnetic transition (MMT), can be independently determined to be $B_m = 20\,000$ Oe by the peak in $dM/dH$ curve (electronic supplementary material figure S3). The complex magnetic behaviour of compound **1** at low temperature and the strain-induced pyroelectric and ferroelectric at room temperature suggest that **1** may have magnetoelectric effect at low temperature.

# 4. Conclusion

In conclusion, strain-induced pyroelectricity and ferroelectricity were observed in single crystal of (4-(Aminomethyl)pyridinium)$_2$MnCl$_4 \cdot$2H$_2$O. There is a magnetic anomaly around 2.5 K. Below 2.5 K, an antiferromagnetic ordering phase appears. In the antiferromagnetic phase, a field-induced spin-flop transition is observed.

Data accessibility. Our data are deposited at the Dryad Digital Repository: https://dx.doi.org/10.5061/dryad.vmcvdncpt [39].

Authors' contributions. K.G., C.L., K.W. and W.Z. grew the single crystals, took the pyroelectric, ferroelectric and DSC measurements., W.L. took the single crystal XRD measurements. K.G. designed the experiments and wrote the paper. All authors participated in the discussion of the article.

Competing interests. We declare we have no competing interests.

Funding. This work was supported by the Natural Science Foundation of Jiangsu Province, China (grant no. BK20170482).

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
