## [Reviewer comments · Royal Society Open Science]

Review History

RSOS-200271.R0 (Original submission)

Review form: Reviewer 1

Is the manuscript scientifically sound in its present form?

Yes

Are the interpretations and conclusions justified by the results?

Yes

Is the language acceptable?

Yes

Do you have any ethical concerns with this paper?

No

Have you any concerns about statistical analyses in this paper?

No

Recommendation?

Major revision is needed (please make suggestions in comments)

Comments to the Author(s)

In this work, the authors reported an Mn-based hybrid compound (4-(Aminomethyl)pyridinium)₂MnCl₄·2H₂O with a P-1 space group, showing small pyroelectric current, as well as a ferroelectric hysteresis loop. The authors claimed that the pyroelectric and the ferroelectric property was attributed to the strain caused by defects. Moreover, temperature-dependent magnetic curves and the M-H curve shows an antiferromagnetic ordering behavior below 2.5 K and A field-induced spin-flop is observed in the antiferromagnetic ordering state. I believe the article passes the basic requirements for publication. There are, however, a number of changes that should be made to the article before publication to address a number of issues with the current version of the manuscript.

1. The introduction part needs to be reorganized, and the current version is poorly written.
2. The authors found that the title compound crystallizes in a centrosymmetric space group, but with pyroelectric and ferroelectric properties. The authors, however, infer only through guesswork that it is due to a stress defect in the crystal. This is not necessarily rigorous, and I strongly suggest that the author prove his conclusion by means of experiments.
3. PXRD pattern should be retested and the baseline should be recalibrated.

Review form: Reviewer 2

Is the manuscript scientifically sound in its present form?

Yes

Are the interpretations and conclusions justified by the results?

Yes

Is the language acceptable?

Yes

Do you have any ethical concerns with this paper?

No

Have you any concerns about statistical analyses in this paper?

Yes

Recommendation?

Major revision is needed (please make suggestions in comments)

Comments to the Author(s)

1. The section of Materials and Methods is over simplified. Please include more details of how the experiment is conducted. Schematic illustration might be helpful.
2. XRD results should be included in the main content as an inset picture of the Figure 1.
3. The author mentioned 'The ferroelectric and pyroelectric behavior in centrosymmetric compound 1 is similar to that in SrTiO₃'. Is there any figure of merit in terms of pyroelectricity to compare the performance of the two materials?
4. In the experimental section, the author indicated that 'a 1047 nm pulsed laser with a power of 100 mW' was used. Is it possible to choose other laser settings?
5. In figure 5, the author provided data of 2K, 5K, 50K and 300K. However, it is still not sufficient to see the trend given that only four temperatures are chosen. So, it should be more direct to catch the trend for readers if there are more measurements for other temperatures.
6. Scientifically, what is the application for the mentioned properties of this material? Is there any advantages of this material over other options?

Decision letter (RSOS-200271.R0)

30-Mar-2020

Dear Dr Gao:

Title: Pyroelectricity and field induced spin-flop in (4-(Aminomethyl)pyridinium)₂MnCl₄•2H₂O
Manuscript ID: RSOS-200271

The editor assigned to your manuscript has now received comments from reviewers. We would like you to revise your paper in accordance with the referee and Subject Editor suggestions which can be found below (not including confidential reports to the Editor). Please note this decision does not guarantee eventual acceptance.

Please submit your revised paper before 22-Apr-2020. Please note that the revision deadline will expire at 00.00am on this date. If we do not hear from you within this time then it will be assumed that the paper has been withdrawn. In exceptional circumstances, extensions may be possible if agreed with the Editorial Office in advance. We do not allow multiple rounds of revision so we urge you to make every effort to fully address all of the comments at this stage. If deemed necessary by the Editors, your manuscript will be sent back to one or more of the original reviewers for assessment. If the original reviewers are not available we may invite new reviewers.

RSC Associate Editor:
Comments to the Author:
(There are no comments.)

RSC Subject Editor:
Comments to the Author:
(There are no comments.)

Reviewers' Comments to Author:
Reviewer: 1

Comments to the Author(s)

In this work, the authors reported an Mn-based hybrid compound (4-(Aminomethyl)pyridinium)₂MnCl₄ · 2H₂O with a P-1 space group, showing small pyroelectric current, as well as a ferroelectric hysteresis loop. The authors claimed that the pyroelectric and the ferroelectric property was attributed to the strain caused by defects. Moreover, temperature-dependent magnetic curves and the M-H curve shows an antiferromagnetic ordering behavior below 2.5 K and A field-induced spin-flop is observed in the antiferromagnetic ordering state. I believe the article passes the basic requirements for publication. There are, however, a number of changes that should be made to the article before publication to address a number of issues with the current version of the manuscript.

1. The introduction part needs to be reorganized, and the current version is poorly written.
2. The authors found that the title compound crystallizes in a centrosymmetric space group, but with pyroelectric and ferroelectric properties. The authors, however, infer only through guesswork that it is due to a stress defect in the crystal. This is not necessarily rigorous, and I strongly suggest that the author prove his conclusion by means of experiments.
3. PXRD pattern should be retested and the baseline should be recalibrated.

Reviewer: 2

Comments to the Author(s)

1. The section of Materials and Methods is over simplified. Please include more details of how the experiment is conducted. Schematic illustration might be helpful.
2. XRD results should be included in the main content as an inset picture of the Figure 1.
3. The author mentioned 'The ferroelectric and pyroelectric behavior in centrosymmetric compound 1 is similar to that in SrTiO₃'. Is there any figure of merit in terms of pyroelectricity to compare the performance of the two materials?
4. In the experimental section, the author indicated that 'a 1047 nm pulsed laser with a power of 100 mW' was used. Is it possible to choose other laser settings?
5. In figure 5, the author provided data of 2K, 5K, 50K and 300K. However, it is still not sufficient to see the trend given that only four temperatures are chosen. So, it should be more direct to catch the trend for readers if there are more measurements for other temperatures.
6. Scientifically, what is the application for the mentioned properties of this material? Is there any advantages of this material over other options?

Author's Response to Decision Letter for (RSOS-200271.R0)

See Appendix A.

RSOS-200271.R1 (Revision)

Review form: Reviewer 1

Is the manuscript scientifically sound in its present form?

Yes

Are the interpretations and conclusions justified by the results?

Yes

Is the language acceptable?

Yes

Do you have any ethical concerns with this paper?

No

Have you any concerns about statistical analyses in this paper?

No

Recommendation?

Accept as is

Comments to the Author(s)

The current version is suitable for publication.

Review form: Reviewer 2

Is the manuscript scientifically sound in its present form?

Yes

Are the interpretations and conclusions justified by the results?

Yes

Is the language acceptable?

Yes

Do you have any ethical concerns with this paper?

No

Have you any concerns about statistical analyses in this paper?

No

Recommendation?

Accept as is

Comments to the Author(s)

The authors have addressed my comments.

Decision letter (RSOS-200271.R1)

Dear Dr Gao:

Title: Pyroelectricity and field induced spin-flop in (4-(Aminomethyl)pyridinium)₂MnCl₄•2H₂O
Manuscript ID: RSOS-200271.R1

It is a pleasure to accept your manuscript in its current form for publication in Royal Society Open Science. The chemistry content of Royal Society Open Science is published in collaboration with the Royal Society of Chemistry.

RSC Associate Editor:
Comments to the Author:
(There are no comments.)

RSC Subject Editor:
Comments to the Author:
(There are no comments.)

Reviewer(s)' Comments to Author:
Reviewer: 1

Comments to the Author(s)
The current version is suitable for publication.

Reviewer: 2
Comments to the Author(s)
The authors have addressed my comments.

Appendix A

Dear Editors and Reviewers:

Thank you for your letter and for the reviewers' comments concerning our manuscript entitled "Pyroelectricity and field induced spin-flop in (4-(Aminomethyl)pyridinium)₂MnCl₄•2H₂O" (ID: RSOS-200271). Those comments are all valuable and very helpful for revising and improving our paper, as well as the important guiding significance to our researches. We have studied the comments carefully and have made corrections which we hope meet with approval. Revised portions are marked in red in the paper. The main corrections in the paper and the responds to the reviewer's comments are as following:

Reviewer: 1

Comment 1: The introduction part needs to be reorganized, and the current version is poorly written.

Response: The introduction part has been reorganized as: "Organic-inorganic hybrid perovskites have attracted great attention due to their rich and excellent physical properties(1-5). In hybrid perovskites crystals, BX₆ octahedra play an important role in their multifunction property. Many ferroelectric materials have been found among organic-inorganic hybrid perovskite materials(4, 6-19). The octahedral structure is important for the ferroelectricity of hybrid perovskites. Some of the organic-inorganic hybrid perovskites are found to possess large spontaneous polarization and large piezoelectric coefficient that is comparable to those of inorganic materials. The excellent ferroelectric property and piezoelectric property of organic-inorganic perovskites make molecular based ferroelectrics become attractive in the field of application(20-22). Here we report a compound (4-(Aminomethyl)pyridinium)₂MnCl₄•2H₂O which have organic-inorganic hybrid perovskite structure. Pyroelectricity and field induced spin-flop was found in the single crystal of compound **1**."

Flexoelectricity reflects a coupling between polarization and strain(23-26). Most materials have flexoelectric effect, in which a strain gradient could induce a polarization and a piezoelectric composite containing no piezoelectric elements. In centrosymmetric materials, a strain gradient can break the inversion symmetry. The strain induced polarization in SrTiO₃ is widely studied. The crystal structure of SrTiO₃ belong to the simple cubic centrosymmetric lattices and the dielectric permittivities of SrTiO₃ is large. The ABX₃ crystal structure and the large dielectric permittivities make it easy for strain to induce a polarization in the SrTiO₃ single crystal(27, 28). Bulk photovoltaic bulk effect can also exist in SrTiO₃, which is caused by flexoelectricity⁴. Pyroelectricity in **1** is similar that found in SrTiO₃(29), which is not caused by the ferroelectricity or spontaneous polarization. The space group of single crystal of **1** is P $\bar{1}$, which is centrosymmetric symmetry. However, small pyroelectric current is detected in **1** using Chnowth technology. Flexoelectric effect may play an important role in the pyroelectric effect in **1**."

Comment 2: The authors found that the title compound crystallizes in a centrosymmetric space group, but with pyroelectric and ferroelectric properties. The authors, however, infer only through guesswork that it is due to a stress defect in the crystal. This is not necessarily rigorous, and I strongly suggest that the author prove his conclusion by means of experiments.

Response: We also want to prove the pyroelectricity of the compound come from stress defect in the crystal. We try to find groups to test flexoelectric effect on the single crystals of the compound. But till now, we don't find such group who can help us. Moreover, the mechanism here will be more complicated than inorganic compounds (SrTiO₃) due to the presence of crystallized water and organic molecules. Anyway, we will go on studying the issue in the future.

Comment 3: PXRD pattern should be retested and the baseline should be recalibrated.

Response: The powder used for PXRD test come from grinding of single crystals of (4-(Aminomethyl)pyridinium)2MnCl₄·2H₂O. As the compound has crystallized water, grinding destroys the crystal structure of the compound. After grinding, the crystallinity of the crystal deteriorates, and amorphization occurs. Therefore, an amorphous packet appears at about 20 degrees of 2θ.

Reviewer: 2

Comment 1: The section of Materials and Methods is over simplified. Please include more details of how the experiment is conducted. Schematic illustration might be helpful.

Response: Details have been added in the materials and methods part.

“Measurements:

The single crystal X-ray diffraction was performed on a Bruker D8 QUEST at room temperature. The Powder X-ray diffraction (PXRD) was measured on a Rigaku D/MAX 2000 PC X-ray diffractometer. DSC measurements of single crystals were recorded by a NETZSCH DSC 200F3. The DSC of **1** was measured from **120 K to 470 K with sweeping speed of 10 K/min**. P–E hysteresis loops were recorded on a Precision Premier II (Radiant Technologies, Inc.). The PE hysteresis loop of **1** was measured on the single crystals at room temperature with frequency of 50 Hz. The dynamic pyroelectric current was measured using the Chynoweth technique. The current was recorded with an electrometer (Keithley 6517B). A 1047 nm pulsed laser with a power of 100 mW was used to heating the sample period. Magnetic measurements of the samples were performed on a Quantum Design SQUID (MPMS XL) magnetometer. The magnetic susceptibility of **1** was measured from 300 K to 2 K with a speed of 5 K/min. “

Comment 2: XRD results should be included in the main content as an inset picture of the Figure 1.

Response: XRD results has been included in Fig.1.

Fig. 1 (a) The crystal structure of 1 at room temperature. (b) Powder XRD pattern of the compound 1 compared with the calculated PXRD pattern deduced from the crystal structure.

Comment 3: The author mentioned ‘The ferroelectric and pyroelectric behavior in centrosymmetric compound 1 is similar to that in SrTiO₃.’. Is there any figure of merit in terms of pyroelectricity to compare the performance of the two materials?

Response: Pyroelectric current was measured on SrTiO₃ bulk by Elena Meirzadeh et. al. (Adv. Mater. 2019, 1904733). As the dynamic pyroelectric method is hard to be used to quantitative characterize of pyroelectric performance, we cannot compare the pyroelectric properties. But both have pyroelectric effect and belong to central symmetric space group.

Figure 1. Pyroelectric measurements. a) Schematic illustration of the measurement setup where SrTiO₃ is subjected to periodic heating pulses using an infrared laser while the current between a top and bottom electrode is measured. b) Pyroelectric current generated by the cycling temperature changes close to room temperature. The insets depict the change in the surface polarization and the current generated upon temperature change. c) A pyroelectric current induced by 0.12 ms laser pulse with 0.88 ms cooling time for allowing the sample to return to its initial stage at the end of each heating and cooling cycle. $\alpha\delta$ can be calculated from the fit to $I \propto 1/\sqrt{t}$ (green) (see Section S2 in the Supporting Information).

Comment 4: In the experimental section, the author indicated that ‘a 1047 nm pulsed laser with a power of 100 mW’ was used. Is it possible to choose other laser settings?

Response: The infrared laser used here just act as pulse heating source. So the laser setting can be changed just making sure that the laser can heat the samples effectively and no other effect can be caused by the laser.

Comment 5: In figure 5, the author provided data of 2K, 5K, 50K and 300K. However, it is still not sufficient to see the trend given that only four temperatures are chosen. So, it should be more direct to catch the trend for readers if there are more measurements for other temperatures.

Response: Above 5K, there’s no special magnetic behavior just ordinary temperature-dependent magnetic ordering. The magnetic ordering trend above 5 K can be seen from Fig. 4.

Fig. 4 The temperature dependent magnetic susceptibility.

Comment 6: Scientifically, what is the application for the mentioned properties of this material? Is there any advantages of this material over other options?

Response: Strain induced pyroelectricity belong to flexoelectric effect. The absence of symmetry constraint makes the flexoelectric materials suitable for most cases

where non-uniform electric field distribution and non-uniform strain distribution exist. The recent discoveries utilized the flexoelectricity into many important application fields such as sensor and actuator, charge transportation, defect formation, domain tailoring, and some open applications like flexo-photovoltaic effect and flexo-caloric effect have been commented. I think now this material don't have advantages over other options. This research can just provide a direction to find more and better flexoelectric materials.